# Low-Profile Dual-Band Reflector Antenna for High-Frequency Applications

**DOI:** 10.3390/s23135781

**Published:** 2023-06-21

**Authors:** Senlin Lu, Shi-Wei Qu

**Affiliations:** School of Electronic Science and Engineering, University of Electronic Science and Technology of China (UESTC), Chengdu 611731, China; 201911022326@std.uestc.edu.cn

**Keywords:** dual band, reflector antenna, low profile, reflectarray, high gain, low cost

## Abstract

A high-gain low-profile reflector antenna with dual-band radiation ability is presented in this paper. The antenna achieves a relative 2 dB gain bandwidth of 10% around *f_l_*, and a relative 2 dB gain bandwidth of 20%, around *f_h_*, where *f_l_* and *f_h_* are the center operating frequencies of the frequency bands of 29.4~32.4 GHz and 142~174 GHz, respectively. To achieve the dual-band radiation ability, a composite dual-band feed with an *f_h_*/*f_l_* ratio of around 5 is proposed as the feed for the reflector antenna, which includes a higher-band circular waveguide and a lower-band coaxial horn. The metallic elliptical surface serves as the subreflector (SR) in the higher band, while the SR is the planar reflectarray in the lower band. Due to the design of the dual reflector, the dual-band reflector antenna features a low focal-to-diameter (F/D) ratio of approximately 0.2. According to the simulated results, the proposed reflector antenna achieves efficiencies of 59.0% and 42.9% at *f_l_* and *f_h_*, respectively. For verification purposes, a Ku/E-band scaled prototype is manufactured. The measured VSWRs, radiation patterns, and gains are in reasonable agreement with the simulated ones, proving the correctness of the proposed design method.

## 1. Introduction

The millimeter wave (MMW) and terahertz (THz) bands are widely used in wireless communications due to their high data rate, wide transmission bandwidth, and high security. Owing to the massive increase in application scenarios, spectrum resources in different frequency bands have been explored and utilized, e.g., the 220 GHz antenna for sensing [1] and 0.3 THz and 0.5 THz antennas for high-speed wireless data transmission [2,3,4]. With the increasing exploitation of spectrum resources, the number of required antennas is also increasing, leading to higher costs. Therefore, a single antenna that can work in different frequency bands can reduce the number of antennas and save on costs. Moreover, in order to guarantee the wireless link budget, it is important to enhance the antenna directivity.

With the development of wireless communication systems, characteristics such as high gain, high efficiency, low profile, and low cost are required for MMW/THz communication systems. In recent decades, researchers have developed various types of antennas to accommodate these requirements. For instance, substrate-integrated waveguides (SIWs) [5,6,7] are employed to achieve high gain. However, these array antennas exhibit low-efficiency performance due to dielectric loss when the array scale increases to 32 × 32. In addition, waveguide slot array antennas in [8,9] are among the solutions for MMW/THz antennas. However, the processing cost is increased due to their intricate structure. To accommodate high-gain and high-efficiency requirements, reflector antennas have been extensively investigated in communication systems.

In recent decades, various methods of designing dual-band reflector antennas have been proposed [10,11,12,13,14,15,16,17,18,19,20,21,22,23,24,25,26,27,28]. Ultra-wideband feed antennas, capable of covering two different bands, have been proposed, e.g., quadridge horn [14], sinuous feed antenna [15], and Eleven feed antenna [16]. However, these antennas have some shortcomings. For example, the phase center of the quadridge horn and the polarization direction of the sinuous feed change based on frequency, which can cause deterioration in the dual-band performance of the reflector antenna. Similarly, the radiation patterns of Eleven feed antenna change significantly based on frequency. Therefore, it cannot be guaranteed that the efficiency of the reflector antenna will maintain an acceptable level throughout the entire operating bandwidth [17].

Additionally, a dual-band feed antenna can be used to illuminate the main reflector antenna to achieve dual-band functionality [18,19,20,21,22]. However, when this approach is applied in MMW/THz bands, dual-band feeds may confront processing challenges due to the complicated structures of the feed antenna [18,19]. To obtain a high aperture efficiency of the reflector antenna, a high focal-to-diameter (F/D) [20,21,22] ratio is required. Generally, the profile of the reflector antenna is relatively high, which contradicts the low-profile requirement.

Essentially, the two methods, i.e., utilizing ultra-wideband feed antennas and using dual-band feed antennas for dual-band operation, focus on the designs of feed antennas. To obtain dual-band capability through the subreflector (SR) of the antenna, a frequency-selective SR (FSSR), which can separate the higher- and lower-band feeds, is introduced [23,24,25,26,27]. Generally, the FSSR is designed to reflect signals in one frequency band and propagate signals in the other frequency band. However, the unavoidable deterioration of radiation occurs due to the imperfect reflection and transmission of the FSSR [26,27], and the use of the FSSR also leads to a high profile.

To achieve a lower profile, dual reflectors are employed in the design, instead of a single reflector. For example, a subreflectarray (SRA) is employed in [28] as the SR which can work in two different frequency bands. To achieve dual-band radiation capability, the elements of the SRA must function well in both of the bands. However, due to limitations on the operating frequency bandwidth of the elements, the application of the SRA is limited, especially when the dual-band reflector antennas operate with a large frequency ratio.

In this paper, a high-gain low-profile reflector antenna with dual-band radiation capability is proposed. A composite dual-band feed with an *f_h_*/*f_l_* of around 5 is utilized as the feed for the reflector antenna, where *f_l_* and *f_h_* are 30.9 GHz and 158 GHz, respectively. Two SRs are designed separately, one is the SRA, employed for lower-band operation, and the other is the metal SR, employed for higher-band operation. Compared to conventional dual-band reflector antennas, the proposed dual-band reflector antenna features a lower profile. Furthermore, the proposed reflector antenna is fabricated, and the measured results verify the simulated dual-band performance in the two bands.

## 2. Antenna Configuration and Design

### 2.1. Main Reflector, SRA, and Higher-Band SR

The configuration of the low-profile dual-band reflector antenna is shown in Figure 1. It consists of a main reflector, a composite feed, an SRA, and a higher-band SR. In detail, the main reflector, which is a ring-focus reflector, is shaped by rotating the curve along the ring-focus curve. The diameter of the main reflector is 78.5*λ_h_*, where *λ_h_* is the free-space wavelength at the higher frequency *f_h_*. Additionally, the composite feed, with the *f_h_*/*f_l_* ratio around 5, consists of a higher-band coaxial horn and a lower-band ring focus. Furthermore, the lower-band ring-focus SRA is a small reflectarray and designed by employing the rules mentioned in [28], while the higher-band SR is a metallic curving SR and formed by rotating the ellipse curve along the Z axis. Moreover, a metal clad layer serves as the SR in the higher band, which is attached to the top surface of the higher-band subreflector. The material of the substrates in the antenna is employed with a relative permittivity of 2.2 and a dielectric loss tangent of 0.01.

The cross section of the geometry is illustrated in Figure 2. The point F_1_ under SR is not only the phase center of the ring-focus main reflector but also one of the focal points for both lower-band and higher-band rays. The point F_2_ is the phase center of the lower-band feed and the other focal point for lower-band rays. The phase center of the higher-band feed coincides with the other focus of the SR, as represented by the point F_3_, which is located on the symmetry axis of the SR. Figure 2 shows the sketch of ray paths in both bands. Different SRs are designed separately to achieve dual-band operation ability and the SR designs, for both bands, are demonstrated as follows:(1)According to the geometric optics (GO) theory, lower-band rays emerging from the point F_2_ are first reflected by the SRA. Then, the rays reach the main reflector. Finally, the rays are reflected into the airspace along the Z axis. The dashed line in Figure 2 represents the virtual ray path from the SRA to the focal point F_1_, which the ray does not actually pass through. The SR can provide the compensation phase for lower-band rays and, as a result, the virtual rays arriving at focal point F_1_ are in phase. Based on the different lengths of lower-band ray paths, the desired compensation phase for SRA elements in various locations can be calculated.

According to the GO theory analysis, the desired compensation phase for each element on the SRA aperture can be calculated by
(1)$$phase_{mn}^{desired}=kR_{1mn}+kR_{2mn}$$
where *k* is the wavenumber in free space, and the spatial distance between the point F_2_ and the *mn*th element of SRA is represented by *R*_1*mn*_. Moreover, the distance between the point F_1_ and the *mn*th element of SRA is represented by *R*_2*mn*_. The SRA is a symmetric construction, and the distance between the center of the SRA and F_2_ is 6.02*λ_h_*. The distance between F_1_ and the surface of the SRA is 1.57*λ_h_*. Moreover, the diameter of the SRA is 15.7*λ_h_*, which is a compromise between the smaller blockage of the SRA and the reduced spillover loss of the lower-band feed.

(2)Based on the GO theory, higher-band rays that emerge at point F_3_ are first reflected by the SR. Then, these higher-band rays pass through point F_1_ and reach the main reflector. Finally, the higher-band rays are reflected into the airspace. Here, *D_S_* is the diameter of the SR. *θ_m_* is the semi-flare angle of the dielectric guide and *F* is the focal length of the main reflector. According to these three parameters and *D*, the curve equations of the main reflector and SR are uniquely determined.

Because the *D_S_* is much smaller than the *D*, the shadow influence of the higher-band subreflector can be ignored and the lower-band SR can be designed without considering the influence of the higher-band subreflector and support dielectric.

### 2.2. Composite Feed

A composite feed operating in MMW/THz bands is proposed. As shown in Figure 3, the composite feed consists of a higher-band feed and a lower-band feed. The higher-band feed is a circular waveguide, while the lower-band feed is composed of a dielectric, a conical coaxial horn, and a waveguide connector. The dielectric serves as a supporting structure that maintains the relative position of the conical coaxial horn and the circular waveguide stable. Due to its small size, the dielectric has minimal influence on the lower-band feeding.

The diameters of circular waveguides are important for the working band. Therefore, it is crucial to determine the dimensions of the waveguides first. Firstly, the diameter of the circular waveguide for the higher-band feed is chosen to be *r*_3_ = 0.68*λ_h_*, ensuring that the TE_11_-mode electromagnetic waves propagate within the frequency range of 0.90~1.10 *f_h_*, as shown in Figure 4b. Secondly, the diameter of the coaxial waveguide for the lower-band feed is chosen to be *r*_4_ = 3.22*λ_h_*, which can sustain the propagation of the first higher-order TE_11_ mode within the frequency range of 0.91~1.10 *f_l_*. Figure 4a shows the transmission coefficients of the first and second higher-order modes of electromagnetic waves in the lower band, propagating in the coaxial waveguide. As a result, the desired first higher-order TE_11_ mode of electromagnetic waves can be propagated in the coaxial waveguide. Thirdly, the waveguide connector is chosen as the mode converter. The first higher-order TE_11_ mode of the coaxial waveguide can be excited by the TE_11_-mode transmission mode of the waveguide connector, as the electric field distribution of the TE_11_-mode is similar between rectangular waveguides and coaxial waveguides. Finally, port 1 of the coaxial waveguide is connected to the rectangular waveguide, as shown in Figure 4b. The figure displays the TE_11_-mode transmission coefficients of the circular waveguide and the coaxial waveguide within the higher and lower bands separately. Therefore, the TE_11_ mode of a coaxial waveguide can be excited by a rectangular waveguide with low loss in the lower band. The TE_11_-mode transmission coefficients of the coaxial waveguide in the lower band can be optimized by the parameters of the rectangular waveguide connector, as shown in Figure 3a. The parameters of the dielectric can be adjusted properly to achieve an acceptable matching impedance in the lower band, and the diameters of the stepped cylinders located at the bottom of the dielectric can be adjusted properly to reduce the feed mismatch loss in the higher band. The simulated reflection coefficients of the composite feed are shown in Figure 5, suggesting that the simulated reflection coefficients are below −15 dB in both bands.

The simulated gain and aperture efficiency in the higher band are shown in Figure 6, based on the full-wave simulations. As a result, the reflector antenna achieves a gain higher than 43 dBi and an aperture efficiency higher than 39% over the entire higher band. Meanwhile, a relative 2 dB gain bandwidth of the higher operating band is approximately 20%, around *f_h_*. Furthermore, the simulated normalized co-polarization (Co-pol) and cross-polarization (Cr-pol) radiation patterns at *f_h_* are shown in Figure 6b. The simulated Cr-pol patterns are −50 dB lower than the maximum direction in both E- and H-planes.

Table 1 summarizes the estimated aperture efficiency and gain loss reduction attributed to *f_h_*. The calculated total losses at *f_h_* are 3.7 dB, with taper loss, spillover loss, and dielectric loss being the main contributing factors. Additionally, the shade loss results from the shielding of the lower-band SR at *f_h_*. Furthermore, the simulated gain closely matches the calculated gain.

### 2.3. Element of Lower-Band SRA

Due to the small diameter of the SRA, a subwavelength element is employed as the element of the SRA. It can more accurately achieve phase compensation than a half-wavelength element in such a small area. The SRA element consists of a cross patch and a rectangular loop [28], as shown in Figure 7. In addition, the size length of the square element is *L* = 0.23*λ_l_*, where *λ_l_* is the free-space wavelength at *f_l_*. The element is etched on a 0.077*λ_l_* thick dielectric substrate with a relative permittivity of 2.2, and a dielectric loss tangent of 0.0009. The reflection phase curves consider different polarizations of the incident wave. The infinite periodic element model is built to simulate the reflection phase at *f_l_*. Moreover, another important consideration in the element analysis is the reflection phase under oblique incidences, as shown in Figure 7. Based on the relative position between the SRA and the lower-band feed, the maximum incident angle is about 50°. Then, a max phase range over of 430° is obtained under different incident angles. It can be seen that the reflection phase of the SRA element varies with the incident angle, and the maximum phase change within the oblique incidence range of 0° to 50° is approximately 80°.

The range of the desired compensation phase is from 0° to 950° according to Formula (1). It is difficult for the SRA element to achieve such a wide compensation phase range from 0° to 950°. Thus, a modified formula is proposed to calculate the desired compensation phase for each element on the SRA aperture:(2)$$phase_{mn}^{desired}=mod(kR_{1mn}+kR_{2mn}+\Delta \Phi ,430)$$

To reduce the zones of sharp phase variation, the desired compensation phase is divided by 430, as the maximum phase range over is over 430°. Moreover, the Δ*Φ* is a constant reference phase that can optimize the desired compensation phase in the SRA. The desired compensation phase, as shown in Figure 8a and represented by Formula (2), occurs when Δ*Φ* is 170°. A large phase variation from 400° to 50° is observed in the middle of the SRA, as illustrated in Figure 8a. This sharp phase variation, approximately 350° between adjacent elements in the middle of the SRA, results in a significant change in the geometrical parameters between adjacent elements. Due to this abrupt variation in the geometrical parameters of adjacent elements, the local periodicity assumption for the elements is not satisfied, which in turn affects the phase shift performance of the elements within the area of sharp variation. Even worse, the area of sharp variation is located in the middle of the SRA, where the energy illuminated from the composite feed is at its strongest. Most of the energy reflected by the SRA is not compensated for the desired phase. Thus, the sharp phase variation arising in the middle of the SRA will lead to a degenerated radiation performance. As shown in Figure 8b, the radiation patterns at *f_l_* show a gain lower than 26 dBi, corresponding to an antenna aperture efficiency lower than 17%. Additionally, the degenerate radiation patterns exhibit high sidelobes.

In order to decrease the influence of sharp phase variation in the SRA, the value of Δ*Φ* is designed to be 90°. It can be clearly seen from Figure 9a that no sharp phase variation arises in the middle of the SRA. The area of unavoidable sharp phase variation is moved to the edges of the SRA, not in the middle area, due to the designed Δ*Φ*. Consequently, the best radiation performance of the reflector antenna is achieved, as shown in Figure 9b. The gain of radiation patterns at *f_l_* is 31.4 dBi, corresponding to an antenna aperture efficiency of 59%. Moreover, the optimized radiation patterns show that the high sidelobes are lower than −13 dB. Table 2 summarizes the estimated aperture efficiency and gain loss reduction attribution at *f_l_*. The calculated total losses at *f_l_* are 2.4 dB, with taper loss as the main loss factor. Shade loss is the shielding of the lower-band SR at *f_l_*. Additionally, the simulated gain is close to the calculated gain.

## 3. Measurement Verifications

To verify the dual-band antenna performance, a prototype was manufactured and measured. Considering that the antenna is composed of metal and linear dielectric materials, excluding ferrites, it meets the fundamental scaling principles versus frequencies. Meanwhile, since the anechoic chamber corresponding to the interesting frequency bands in our university was unavailable, it was difficult to accurately measure the radiation patterns at terahertz frequencies. Therefore, the dual-band MMW/THz antenna was scaled to Ku/E-band, and Ku-band performance was accurately measured. The higher band performance has not been experimentally verified, but the simulated results show that the dual-band antenna in the higher band can achieve high gain and high radiation efficiency. The scaled antenna has an aperture dimension of 300 × 300 × 93 mm^3^, as shown in Figure 10a,b. Meanwhile, the composite feed and SRA are shown in Figure 10c,d. The machining process was employed to fabricate all parts of the antenna except the SRA, and the Printed Circuit Board (PCB) technique was utilized to manufacture the SRA. Because the reflector was assembled from several parts using screws and glue, it was difficult to precisely take account into the errors of assembly when the model was simulated.

The simulated and measured reflection coefficients of the scaled feed in the Ku band are shown in Figure 11. The Ku-band performance of the scaled prototype can represent the low-band performance of the proposed antenna. There are slight differences between the simulated and measured reflection coefficients, mainly due to fabrication tolerance and assembly misalignment. The measured reflection coefficients are below −15 dB within the desired frequency band of 14.7~16.2 GHz.

The gain of the manufactured antenna is shown in Figure 12a. As can be seen, the measured gains are higher than 29.9 dBi and the measured total efficiencies are better than 43% within the frequency band of 14.7~16.2 GHz. Meanwhile, the measured gain results show that the manufactured dual-band reflector antenna has a gain variation of less than 2 dB from 14.7 GHz to 16.2 GHz. The measured peak gain is 31.4 dBi, indicating an antenna aperture efficiency of 60.2% at 15.2 GHz, as shown in Figure 12b. Thus, the proposed dual-band reflector antenna can achieve high aperture efficiency and high gain with a low F/D of about 0.2. Moreover, the difference between the measured and simulated gains is less than 1 dB in the 14.7~16.2 GHz band.

Figure 13 shows the comparison of simulated and measured normalized Co-pol and Cr-pol gain patterns at 15.4 GHz. As shown in Figure 13, the measured first sidelobe levels (SLLs) are below −10 dB in both the E-/H-planes. In terms of Co-pol, the measured main beam and first SLLs show good agreement with the simulated results in both the E-/H-planes. Additionally, the Cr-pol level of the measured results is below −25 dB, which is higher than the simulated results. The performance of the fabricated dual-band reflector antenna is sensitive to the distance between SRA and the conical coaxial horn. The copper clad is attached to the surface of the support substrate, serving as the higher-band SR, and an air gap appears between the surface of the support dielectric and the dielectric in the E-band feed. The difference between the measured and simulated results is probably caused by the air gap. Nevertheless, the agreement of the measurement and the simulation indicates that the proposed dual-band reflector antenna is designed properly.

To clearly demonstrate the good radiation performance of the designed reflector antenna, Table 3 presents a comparison between the proposed dual-band reflector antenna and several reported works on dual-band MMW/THz reflector antennas. Compared to other dual-band MMW/THz antennas, the proposed antenna achieves high gain and high aperture efficiency at a low cost. Meanwhile, the proposed dual-band reflector antenna features a lower profile than that of most reported dual-band reflector antennas.

## 4. Conclusions

In this paper, a dual-band MMW/THz high-gain low-profile reflector is proposed. The composite feed, with an *f_h_*/*f_l_* ratio of around 5, consists of a lower-band coaxial horn and a higher-band ring-focus feed. The lower-band SRA is a small planar reflectarray, while the higher-band SR is an elliptical SR. Due to the SRs, the dual-band reflector antenna can achieve a low F/D (F/D = 0.2) in both bands. To validate the proposed design method, a scaled dual-band reflector antenna is manufactured, assembled, and its lower-band radiation performance is measured. In the lower band, the proposed antenna achieves a 31.4 dBi peak gain, and the aperture efficiency is about 60.2%. Meanwhile, the proposed antenna achieves a 45.0 dBi peak gain in the higher band, and the aperture efficiency is above 44%. When operating in the lower band, the proposed antenna achieves a relative 2 dB gain bandwidth of about 10%, around *f_l_*, while the proposed antenna, when operating in the higher band, achieves a relative 2 dB gain bandwidth of approximately 20%, around *f_h_*. The measured results, including the radiation patterns, gain, and reflection coefficient results, all agree well with the simulations. Hence, the effectiveness of the proposed method in designing a high-gain low-profile reflector antenna with dual-band radiation capability is demonstrated. The antenna has the characteristics of low cost and high gain, providing a solution for commercial applications in dual-band communication.

## Figures and Tables

**Figure 1 sensors-23-05781-f001:**
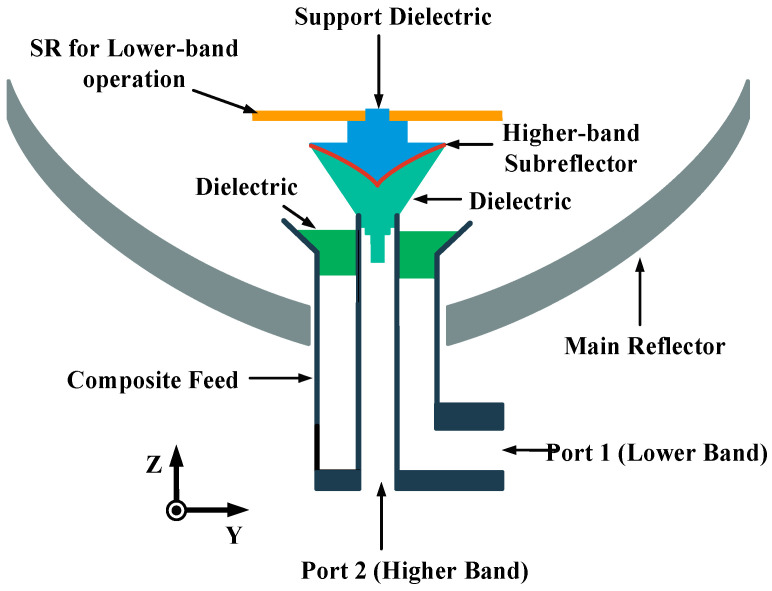
Configuration of the low-profile dual-band reflector antenna.

**Figure 2 sensors-23-05781-f002:**
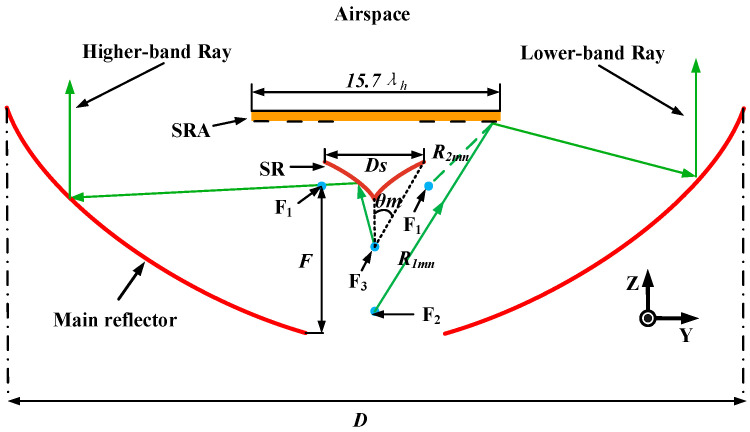
Sketch of ray paths in the lower and higher bands.

**Figure 3 sensors-23-05781-f003:**
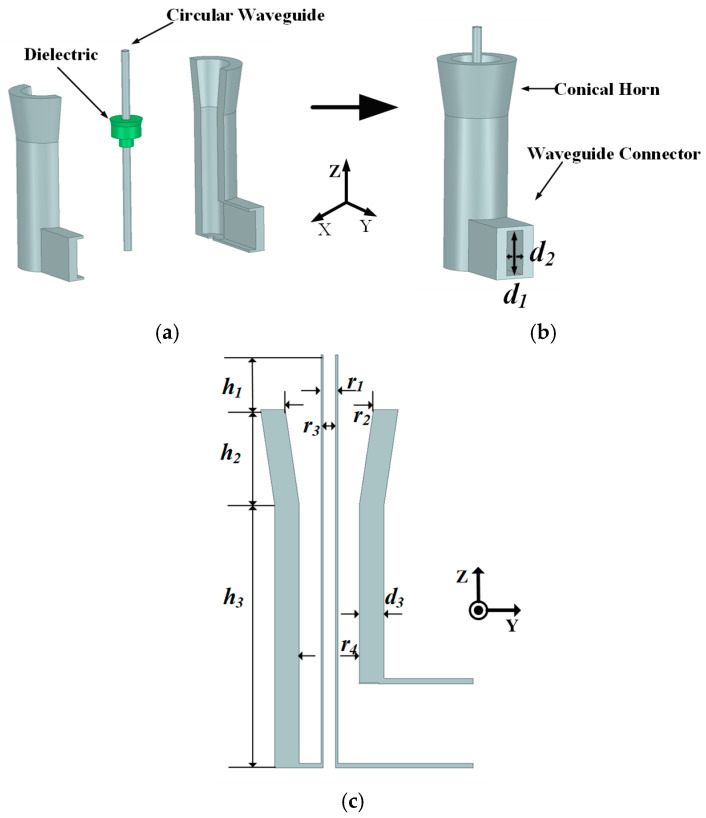
Geometries of (**a**) split view, (**b**) 3D view, and (**c**) cross section of composite feed in YOZ plane. Key parameters: *r*_1_ = 0.88*λ_h_*, *r*_2_ = 4.71*λ_h_*, *r*_3_
*=* 0.68*λ_h_*, *r*_4_ = 3.22*λ_h_*, *d*_1_ = 4.06*λ_h_*, *d*_2_ = 1.68*λ_h_*, *d*_3_ = 1.31*λ_h_*, *h*_1_ = 2.88*λ_h_*, *h*_2_ = 4.97*λ_h_*, *h*_3_ = 15.47*λ_h_*, *D =* 78.5*λ_h_*, *D_s_ =* 2.62*λ_h_*, *θ_m_ =* 45°, *F =* 14.92*λ_h_*.

**Figure 4 sensors-23-05781-f004:**
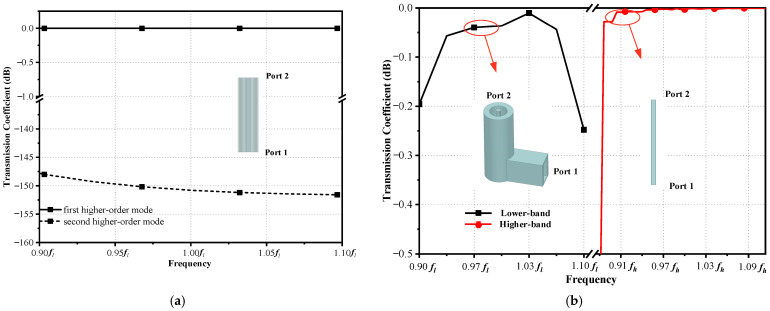
Simulated transmission coefficients of waveguide: (**a**) Coaxial waveguide within lower band. (**b**) Coaxial waveguide with waveguide connector and circular waveguide.

**Figure 5 sensors-23-05781-f005:**
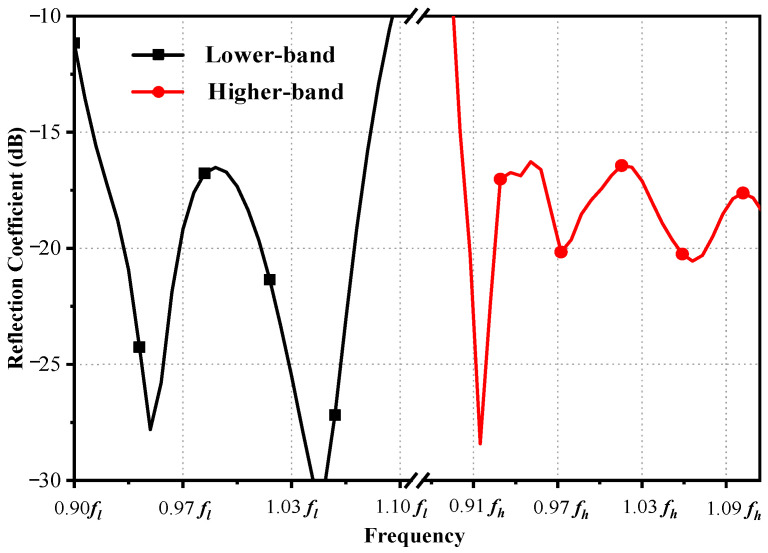
Simulated reflection coefficients of composite feed.

**Figure 6 sensors-23-05781-f006:**
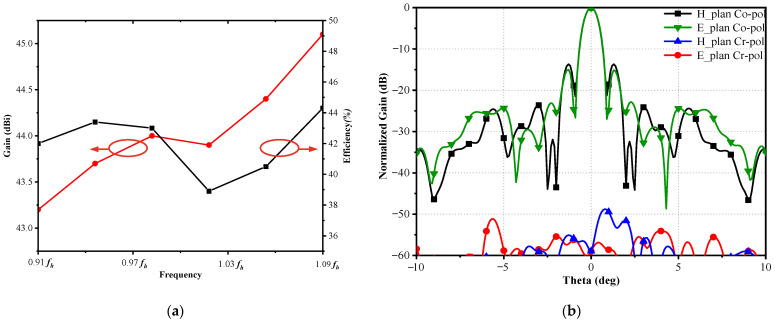
Simulated (**a**) gain and aperture efficiency and (**b**) normalized far-field radiation patterns at *f_h_*.

**Figure 7 sensors-23-05781-f007:**
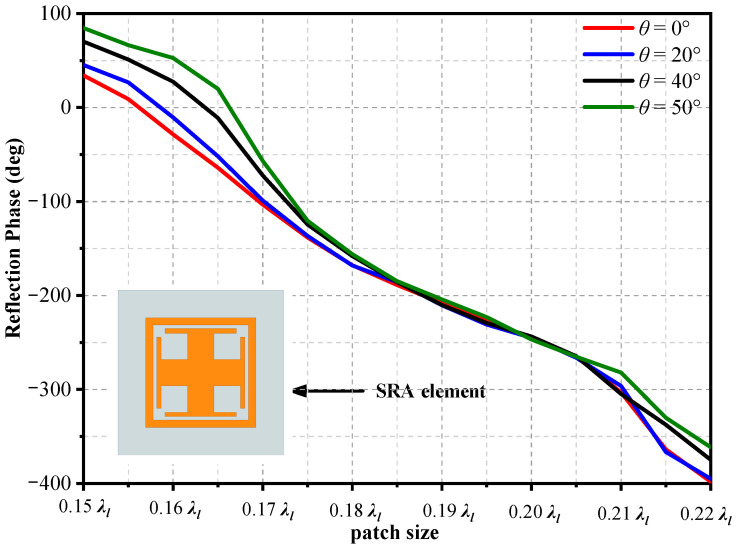
Reflection phase of the element under oblique incidences at *f_l_*.

**Figure 8 sensors-23-05781-f008:**
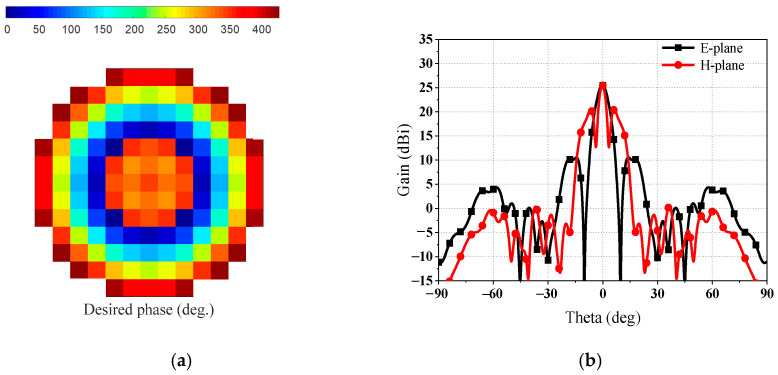
In the condition that the Δ*Φ* is 170°: (**a**) desired compensation phase of SRA and (**b**) radiation patterns at *f_l_*.

**Figure 9 sensors-23-05781-f009:**
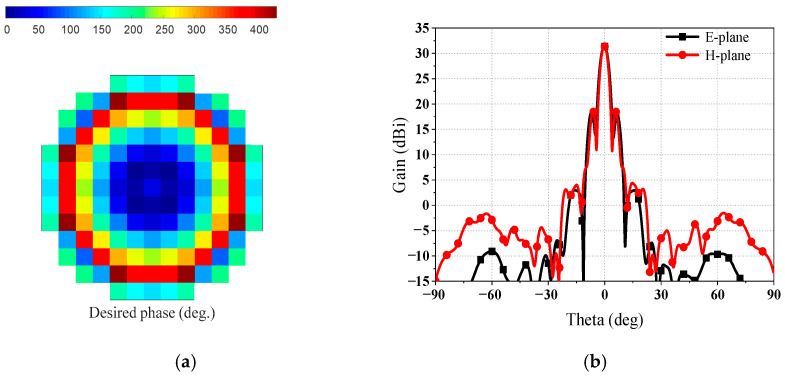
In the condition that the Δ*Φ* is 90°: (**a**) desired compensation phase of SRA and (**b**) radiation patterns at *f_l_*.

**Figure 10 sensors-23-05781-f010:**
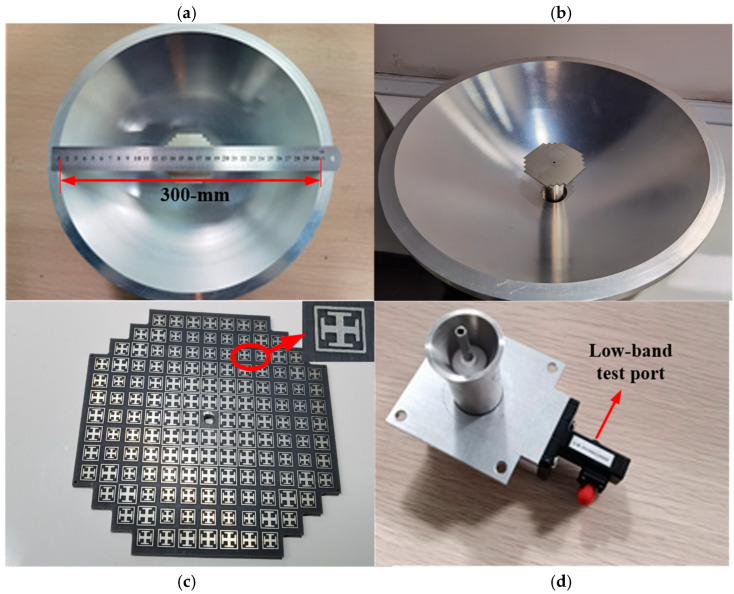
Photographs of the manufactured reflector antenna: (**a**) Top view and (**b**) 3D view of dual-band reflector antenna. (**c**) The SRA. (**d**) The composite feed.

**Figure 11 sensors-23-05781-f011:**
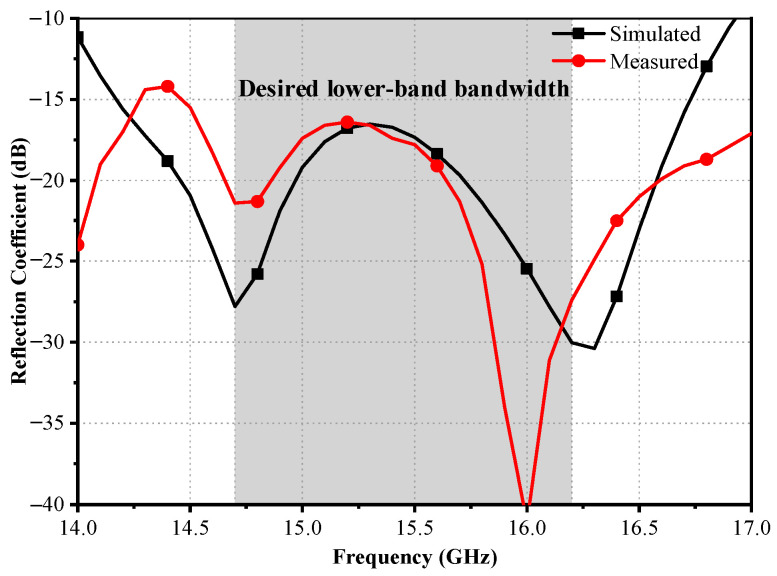
Simulated and measured lower-band reflection coefficients of the manufactured composite feed.

**Figure 12 sensors-23-05781-f012:**
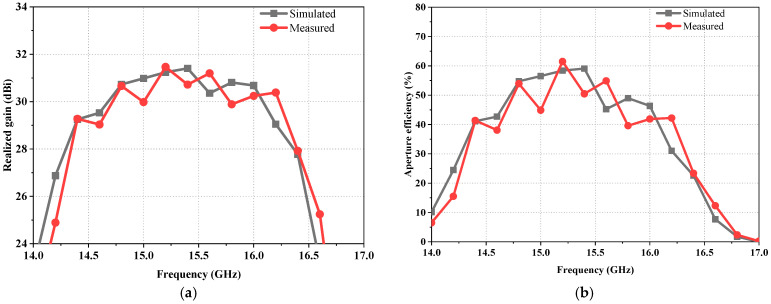
Simulated and measured (**a**) gains and (**b**) aperture efficiency of the reflector antenna within the lower band.

**Figure 13 sensors-23-05781-f013:**
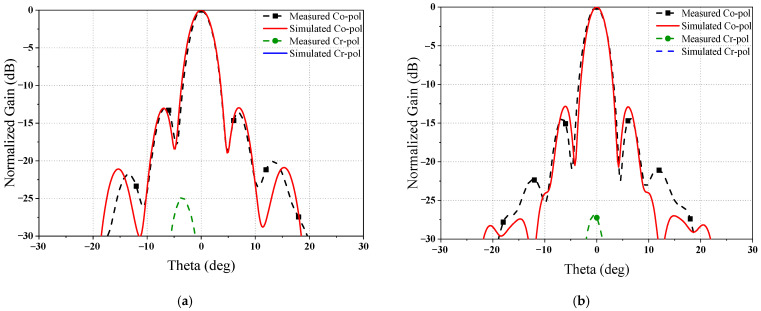
Simulated and measured radiation patterns at 15.4 GHz in normalized (**a**) H-plane and (**b**) E-plane.

**Table 1 sensors-23-05781-t001:** Loss budgets and aperture efficiency at *f_h_*.

Factors	Gain (Efficiency)
Maximum directivity (dB)	47.66 (100%)
Feed mismatch loss (dB)	0.066 (98.5%)
Dielectric loss (dB)	0.97 (78.8%)
Spillover loss (dB)	0.97 (63.0%)
Taper loss (dB)	1.10 (48.5%)
Shade loss (dB)	0.50 (43.2%)
Calculated gain (dB)	43.85 (43.2%)
Simulated gain (dB)	44.0 (42.9%)

**Table 2 sensors-23-05781-t002:** Loss budgets and aperture efficiency at *f_l_*.

Factors	Gain (Efficiency)
Maximum directivity (dB)	33.70 (100%)
Feed mismatch loss (dB)	0.46 (90.5%)
Dielectric loss (dB)	0.066 (89.1%)
Spillover loss (dB)	0.46 (80.2%)
Taper loss (dB)	1.31 (59.3%)
Shade loss (dB)	0.09 (58.2%)
Calculated gain (dB)	31.4 (58.2%)
Simulated gain (dB)	31.4 (59.0%)

**Table 3 sensors-23-05781-t003:** Comparison between references and our work.

Ref	F/D	Frequency Range (GHz)	Peak Gain (Lower/Higher Band)	Efficiency (Higher/Lower Band)	Type *
[28]	0.45	21~25.5/31.5~35.5	34.6 dBi/38.1 dBi	52.6%/54.6%	Horn + SRA
[29]	0.32	3.4~3.8/24.9~36	13.0 dBi/32.1 dBi	42%/47%	Dual-band horn + SR
[30]	0.3	35/94	43.6 dBi/51.3 dBi	26.6%/21.8%	Dual-band horn
[31]	0.5	65~80/115~125	20.3 dBi/21.9 dBi	4.8%/2.7%	Dual-band horn
[32]	1	35.5/94	44.1 dBi/49.6 dBi	40.6%/20.5%	Dual-band horn
[33]	1.95	20~28/65~75	23.2 dBi/30.7 dBi	43.8%/33.7%	Dual-band horn
This work	0.2	29.4~32.4/142~174 **	31.4 dBi/45.0 dBi	59.0%/44.0%	Composite horn + SRA

* Type represents the form of feed and SR. ** The frequency range is the simulated frequency range.

## Data Availability

Not applicable.

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
