# Peer review of "Low-Profile Dual-Band Reflector Antenna for High-Frequency Applications"

_sensors, 2023, doi:10.3390/s23135781_

Round 1

Reviewer 1 Report

The title of the paper does not fit with the work. To be more attractive the authors use mm-wave and THz in the title. However, apart a generic theory the implementation is in microwave band, lower than 30 GHz.

No excuse that you cannot measure. Therefore, do not claim mm-wave or THz.

For which applications these antennas are build? How the frequency were chosen? 

I will agree with the publication only if the authors are totally honest. Do not claim that you made something only to impress the readers and do not lie!!! This is not professional.

Please revise the english when ready.

Reviewer 2 Report

The latest references are not added to the manuscript. References are only up to the year 2021. Compare the results by adding the latest references. also, it contains 26% plagiarism when checked with Turnitin software. Reduce it

Some references are given below and must be added to the manuscript:

1. Sidhu, A. K., & Sivia, J. S. (2023). Design of Wideband Fractal MIMO Antenna using Minkowski and Koch Hybrid Curves on Half Octagonal Radiating Patch with High Isolation and Gain for 5G  Applications. Advanced Electromagnetics12(1), 58–69. https://doi.org/10.7716/aem.v12

2. Pragya, P., Sivia, J.S. Design of Minkowski Curve-Based Slotted Microstrip Patch Antenna Using Artificial Neural Network. J. Inst. Eng. India Ser. B 104, 129–139 (2023). https://doi.org/10.1007/s40031-022-00813-8

3. 

  1. Amandeep Kaur Sidhu & Jagtar Singh Sivia (2022) Design of a novel 5G MIMO antenna with its DGP optimization using PSOGSA, International Journal of Electronics, DOI: 10.1080/00207217.2022.2148288 4. Bhatia, S.S., Sivia, J.S. & Sarao, A.K. Lantern Logo Shaped Novel Monopole Antenna with Semi-Circular Notch Loaded Partial Ground Plane for Ultra-Wideband Wireless Applications. Wireless Pers Commun 126, 3211–3231 (2022). https://doi.org/10.1007/s11277-022-09860-2
  2.  

improve English of the paper

Reviewer 3 Report

11.      Figure 4, for a valid comparison the same scale needs to be used along the vertical axis.

2.      Figures 4 and 5, the frequency is labelled in GHz, while it is presented in terms fl and fh.

3.      It would be useful to mention fl and fh as this will also confirm if this is a wide-band or dual-band antenna.

4.      Figure 6a, please indicate which curve represents the gain and which curve represents the efficiency?  Why are these data presented only for the higher frequency band?

5.      Tables 1 and 2, how the gain was calculated?

6.      It is strange that the authors don’t mention the operating frequency until the reader is more than halfway through the manuscript to realise this is neither a mmWave nor a THz antenna.

7.      From Figure 11, it is obvious that this a wideband antenna instead of a dual band antenna.

88. The authors mention that measurements are presented for the lower band only, is this true for Figure 11 as well? What is the higher X-band frequency range?

Minor revisions would improve the manuscript quality.

Reviewer 4 Report

This manuscript is devoted to development of a low-profile dual-band reflector antenna with high gain. Antenna Configuration and Design are proposed and realized. The results of measuring the characteristics of reflector antenna that proposed and fabricated, and its comparison with the simulation results of dual-band performance in the two bands are presented. Besides the comparison of characteristics of proposed dual-band reflector antenna with results of one previous work of these authors and five works of other scientific groups was carried out. The topic of manuscript is important for scientific groups in area of wireless communication, remote sensing etc.

There are some points to correct or to make the information more clear:

1)      It is necessary to check the text. What means frequency length in phrase “… corresponding to a higher frequency length of 78.5λ”(16th line)? What is “speration” in Fig1? It must be THz in“0.5-Thz” (29th line).  It is better to use the word of “Moreover” instead “What’s more” in “What's more, another important consideration in the element analysis is the reflection phase under oblique incidences, as shown in Fig. 7.”(220th -221st lines).

2)      It is necessary to explain the abbreviation at first mention. Some abbreviation, e.g. GO (for geometric optics possibly) (first mention in 110th line) is without explanation.

3)      It is better to add all values presented in Figure 2 in Figure caption as in Figure 3.

4)      Are the red and black curves in Fig 4b same as in Fig.4a? It is necessary to mark in legend or in Figure caption. This question is concerned also the Fig 5.

5)      What are the red and black curves in Fig 6a? It is not clear in present form. It is necessary to mark that red is gain and black is efficiency (according to text) in Figure caption.

6)      It is necessary to explain the orange figure in grey square in Figure 7 in Figure caption. Is it element of SRA (patch)?

7)      Authors write “the measured reflection coefficients are below -15 dB in frequency band of 14.7 ~ 16.2 GHz”, describing the fig.11, but the red curve is below -15 dB from about 14,5 to 17  GHz. The described above frequency range is included in full frequency range, but if authors describe the results in the Figure, they should explain results in the Figure and highlight the more narrow band if it is necessary.

This manuscript is written sufficiently clear and describes with details the results of development of a low-profile dual-band reflector antenna as well as simulating and measuring its characteristics.

The manuscript can be published after minor revisions.

  It is necessary to check the text. What means frequency length in phrase “… corresponding to a higher frequency length of 78.5λ”(16th line)? What is “speration” in Fig1? It must be THz in“0.5-Thz” (29th line).  It is better to use the word of “Moreover” instead “What’s more” in “What's more, another important consideration in the element analysis is the reflection phase under oblique incidences, as shown in Fig. 7.”(220th -221st lines).

This manuscript is written sufficiently clear and require minor revision.

Round 2

Reviewer 1 Report

OK for now, but the title in my opinion still not adequate, there are not THz experiments in the paper. Even 27 GHz is NOT mm-wave, mm-waves starts at 30 GHz, and even there is exaggerated, wavelength is 10 mm.

Reviewer 3 Report

1.      “It would be useful to mention fl and fh as this will also confirm if this is a wide-band or dual-band antenna.”

Please provide the exact values of fl and fh , in GHz, early in the manuscript for clarity.  The individual bandwidths of the two bands have been merged to create a wide band antenna. If it is a dual band antenna, then please provide the individual bandwidths of each band. 

2.      “It is strange that the authors don’t mention the operating frequency until the reader is more than halfway through the manuscript to realise this is neither a mmWave nor a THz antenna.”

Please avoid the vague comments and identify fl and fh earlier in the manuscript. There is nothing wrong with designing the antenna at higher frequency and build a scaled down prototype to demonstrate the concept provided this is clarified at the beginning of the manuscript.

3.      “From Figure 11, it is obvious that this a wideband antenna instead of a dual band antenna.”

The reflection coefficient is less than -10 dB throughout the whole bandwidth. How you identified the start and end points of the lower band?. Please state clearly in a table the frequency range and percentage bandwidth of each band.
